# Pesticide Residue Trends in Fruits and Vegetables from Farm to Fork in Kampala Metropolitan Area, Uganda—A Mixed Methods Study

**DOI:** 10.3390/ijerph19031350

**Published:** 2022-01-26

**Authors:** Charles Ssemugabo, David Guwatudde, John C. Ssempebwa, Asa Bradman

**Affiliations:** 1Department of Disease Control and Environmental Health, School of Public Health, Makerere University College of Health Sciences, Kampala P.O. Box 7072, Uganda; jssemps@musph.ac.ug; 2Department of Epidemiology and Biostatistics, School of Public Health, Makerere University College of Health Sciences, Kampala P.O. Box 7072, Uganda; dguwatudde@musph.ac.ug; 3Department of Public Health, School of Social Sciences, Humanities and Arts, University of California Merced, Merced, CA 95343, USA; abradman@ucmerced.edu; 4Center for Children’s Environmental Health Research, School of Public Health, University of California, Berkeley, CA 94704, USA

**Keywords:** exposure, handling and processing, stakeholder, supply chain, Uganda

## Abstract

This mixed methods study used laboratory measurements of pesticide residues in produce, semi-structured questionnaires, and in-depth interview data to describe trends in pesticide residue in produce and handling and processing practices for fruits (watermelon and passion fruit) and vegetables (tomato, cabbage, and eggplant) along the farm to fork chain. Of the 50 farmers visited, 34 (68.0%) sold their fruits and vegetables to transporters, 11 (22.0%) to market vendors, and 4 (8.0%) directly to homes and restaurants. The majority 42 (93.3%) of the consumers (home/restaurant) purchased their fruits and vegetables from market vendors and transporters. Washing with water or vinegar, wiping with a cloth, peeling the outer layer, and blending and cooking were the most common post-harvesting processing methods used by stakeholders along the supply chain. Some farmers and market vendors reported spraying fruits and vegetables with pesticides either prior- or post-harvest to increase shelf life. Statistically significant decreasing pesticide residue trends along the farm to fork chain were observed for dioxacarb, likely due to degradation or washing, peeling, cooking, blending, or wiping by consumers. Increasing trends were observed for methidathion and quinalphos possibly due to pesticide applications. There is a need in Uganda to promote practices that minimize pesticide use and exposure through diet, while maintaining food integrity.

## 1. Introduction

Pesticide use has increased globally with increasing demand for food, modernization of agriculture, and pesticide marketing. In Africa, the average amount of pesticides used per hectare increased from 0.32 kg in 1990 to 0.39 kg/ha in 2019 [1]. In Uganda, the average amount of pesticides used per cropland has remained stable at 0.01 kg/ha over the years [1], although pesticide use for agriculture increased from 338 tonnes of active ingredients in the 1960s to 82,560 tonnes in 2019 [2]. The increase in pesticide use results from increased demand for food attributed to population increases from 16.7 million in 1991 to 43 million in 2021 [3]. During pesticide applications, many farmers do not follow recommended mixing concentrations on label instructions [4]. Some farmers also do not comply with the recommended pre-harvest intervals between pesticide applications and harvesting of fruits and vegetables. Additionally, farmers and market vendors use pesticides to increase fruits and vegetables’ shelf life. The presence of high pesticide residues in fruits and vegetables may in part be due to the low levels of knowledge among farmers of good agricultural practices (GAP) and sustainable agriculture methods such as crop rotation and minimal tillage [5]. The Uganda Agricultural Chemical (Control) Act of 2006 lacks clear guidelines on pesticide application practices [6]. In fact, the GAP guidelines developed by the Food and Agricultural Organization (FAO) are minimally enforced by agricultural extension workers [7]. These practices result in increased pesticide residues in fruits and vegetables that leave the farm enroute to the final consumer [8]. These residues, if consumed, can pose acute or chronic risks to human health [9] depending on the level of residue contamination and/or vulnerability of the consumer.

Consumption of pesticide residues in fruits and vegetables can decline along the farm to fork chain. Depending on the physical and chemical properties of a given pesticide, residues may break down over time by hydrolysis, oxidation, microbial degradation, photodegradation, and heat degradation, among other mechanisms [10]. Storage and post-harvest practices in industrial or domestic locations can also affect residues. Domestically, processes such as washing, peeling, comminution (chopping, blanching and crushing), juicing, cooking, milling, and oil production are reported to alter pesticide concentration [10,11]. Post-harvest handling and processing methods are primarily aimed at slowing down physiological processes while increasing the shelf life of fresh produce [12,13,14]. Post-harvest processing can reduce or increase pesticide residue concentration in food [15]. The use of chemicals post-harvest to increase product shelf life has been reported to increase pesticide residue concentration [16]. In summary, at every stage in the food chain, post-harvest handling and processing methods can affect the concentration of pesticide residues.

Given pesticides can be introduced at any stage along the food chain, the farm to fork model provides an opportunity to review all stages including production, distribution, storage and handling from primary production to consumption [17]. This model places the responsibility of food safety on all stakeholders along the food chain [17,18]. An example of this model is *Escherichia coli* (*E. coli*) contamination in salads in Ghana that showed that salads prepared by street vendors contained more *E. coli* than those sampled at the farm, indicating that contamination occurred along the chain [19]. In other cases, some transporters and vendors apply pesticides to increase their shelf life, including spray application and fumigation. In summary, factors along the food chain greatly influence the amount of pesticides that consumers are exposed to. In this study, a mixed-methods concurrent explanatory approach was used to examine pesticide residue trends in produce and handling and processing practices for fruits and vegetables from farm to fork in the Kampala Metropolitan area in Uganda.

## 2. Materials and Methods

We carried out a mixed-methods concurrent explanatory study including sample collection and laboratory analysis of fruits and vegetables, administrating a semi-structured questionnaire, and conducting qualitative in-depth interviews with stakeholders along the produce supply chain in the Kampala Metropolitan Area (KMA) in Uganda. The study was conducted in three of the six districts in the KMA: Wakiso, Kampala and Mukono. The three districts have a population of approximately 10 million people, which is 15% of the total Ugandan population [20], and cover an area of 1000 km^2^ [21]. Agriculture production and trade is the main economic activity within the KMA, supporting approximately 40% of the population [20]. KMA has the largest market for fresh produce in Uganda including markets, restaurants, street vending, and many middle- and high-income homes that consume fruits and vegetables grown within the KMA and in other parts of Uganda.

All study procedures were reviewed and approved by the Makerere University School of Public Health Higher Degrees, Research and Ethics Committee, and registered by the Uganda National Council for Science and Technology (SS 5203). Participation in the study was voluntary and all participants (owners of farms, and restaurants, market managers, street fruit and vegetable vendors and household heads) and stakeholders provided written informed consent. All samples, questionnaires and interviews were coded with an anonymous identification number.

### 2.1. Quantitative Methods

#### 2.1.1. Laboratory Measurements

A total of 160 samples of fruits (watermelon and passion fruit) and vegetables (tomato, cabbage and eggplant) were collected from farms, markets, restaurants and homes—the key stages in the food chain. The detailed methodology used to collect the samples has been described elsewhere [22]. Briefly, working with agricultural extension workers and community leaders, 50 farms were selected—25 in Wakiso district and 25 in Mukono district. Using sales information from the selected farms, 10 community markets were identified to participate in the study. Street vendors, restaurants and homes were selected within communities served by the selected markets with the help of community leaders. Fresh produce samples were collected in sterile polyethylene bags or PET (polyethylene terephthalate) plastic containers. Sampling involved purchasing fruits and vegetables from the selected farms, markets, restaurants, street vendors and homes. A unique identification number was applied to each sample, which were sealed to avoid handling that could affect the integrity of the fruit or vegetable. Samples of ready-to-eat foods were collected from restaurants; juices and salads that do not contain fat-soluble substances were bought from selected restaurants. Through community leaders, homes were identified and asked to prepare samples of fruits and vegetables in the forms they usually consume them, such as juice, salad or sauce. Participating homes were reimbursed based on their estimation of the cost incurred to prepare the sample. Three replicate produce samples were collected at each location measuring at least 1 kg for small or medium fruits and/or vegetables and 2 kg for larger produce as suggested by Codex guidelines [23,24]; prepared food samples were at least 1 kg or 1 L in the case of juice. The samples were stored in a cooler and transported to the Directorate of Government Analytical Laboratory (DGAL) within 8 h. At DGAL, the samples were stored in the freezer at −20 °C until preparation, extraction, and analysis.

The samples of fruits and vegetables and restaurant- and home-prepared food samples were screened for a total of 93 pesticides (Appendix A). The samples were prepared, cleaned and extracted using the Quick, Easy, Cheap, Effective, Rugged and Safe (QuEChERS) approach for determination of pesticide residues [25]. Briefly, 200 g of each sample produce was chopped, grinded and blended to homogenize the sample; homogenization was conducted for 0.5–1 min to avoid enzymatic degradation of the analytes. After homogenization, the samples were put into containers and immediately frozen in order to minimize the risk of degradation of pesticide residues present. Ten grams of homogenized sample was mixed with 3 g of sodium bicarbonate (NaHCO_3_) and 20.0 mL ethyl acetate and centrifuged. The samples were then vortexed for ~30 s and placed on a mechanical shaker at 300 rpm/min for 15 min; and 10 g of anhydrous sodium sulphate (Na_2_SO_4_) was then added and vortexed for 1 min and centrifuged for 3 min. We filtered the crude extract using a 0.2-micron PTFE syringe filter. The final supernatant layer (0.50 g/mL) was transferred into the vials and injected into the Liquid chromatography-Tandem Mass Spectrometer (LC-MS/MS) for analysis of pesticide residues.

Pesticide residue analysis was carried using an Agilent Liquid Chromatography–Tandem Mass Spectrometer (LC-MS/MS) (Agilent Technologies, Santa Clara, CA, USA) system. Chromatographic separation of the targeted analytes was performed using a ZORBAX RRHD Eclipse plus C18 Capillary column with dimensions, 2.1 × 150 mm, 1.8 µm (part number 959759-902) installed on an Agilent 1290 Infinity II LC system with an Agilent 1290 Infinity II high-speed pump (G4220A), Agilent 1290 Infinity II autosampler (G4226A) and Agilent 1290 Infinity II thermostatted column Compartment (G1316C). The LC conditions used were as follows: column temperature was set at 40 °C, injection volume was 5 µL, mobile phase A was 5 mM ammonium formate in water with 0.1% formic acid and mobile phase B was 5 mM ammonium formate in Methanol with 0.1% formic acid. The mobile phase flow rate was 0.3 µL/min. The gradient elution program was 5% B at 0 min, 30% B at 3 min, 100% B at 17 min, 100% B at 20 min and post-run 3 min. The needle flush port was washed for 12 s. An Agilent 6490 triple quadrupole MS system was used for detection of separated analytes with an electrospray ion source (ESI) operated in dynamic Multiple Reaction Monitoring (dMRM) mode. Simultaneous positive/negative ionization mode was used, with nebulizer voltage set at 35 psi, capillary voltage at +(−) 3500 v and drying gas temperature set at 300 °C. The drying gas was nitrogen, whereas the collision cell gas was argon. Agilent MassHunter software version B.08 (Santa Clara, CA, USA) was used.

Pesticide residues were summarized using descriptive statistics (95th interquartile range and maximum values) for all analytes detected (Appendix A). Out of 62 pesticide detected, 21 whose concentrations either increased or decreased along the chain are presented with means and standard deviations (Appendix A) at different stages from farm to fork. Pesticide residue concentrations were expressed in nanograms per kilogram (ng/kg).

#### 2.1.2. Semi-Structured Questionnaire Data

Semi-structured interviews were conducted among 186 stakeholders involved in the post-harvest handling and processing of fruits and vegetables from the farm to the fork. These included: farmers, transporter, market/street vendors, and consumers (restauranteurs/household heads). Farmers from the 50 farms from whom a fruit or a vegetable sample had been collected were interviewed and asked to provide the contact information of the person they sell their produce to. Following up with farm customers, 36 transporters, 54 market vendors, and 46 consumers were identified and interviewed through phone calls.

Data were collected using an interviewer-administered semi-structured questionnaire. The semi-structured questionnaire was developed based on the existing literature [16,26,27,28,29]. Information was collected, including sociodemographics, pesticide use behaviors, and post-harvest handling and processing practices. We used one questionnaire to collect data from all stakeholders involved in each individual chain from farm to fork; however, sociodemographic characteristics were only collected from farmers at the entry point of the supply chain. Phone interviews with the next stakeholders in the fruit and vegetable chain were then conducted. Data were collected by a team of six (6) trained research assistants who were university graduates. The questionnaire was initially pretested in the Jinja district, which is outside the study area and feedback was incorporated to ensure clarity of the tool prior to data collection. Questionnaires were checked by the team leader for inconsistencies and completeness on a daily basis. Where serious discrepancies were found, a call back to the participant was completed.

Data were entered in Epiinfo version 7.2 (Centre for Disease Control, Atlanta, GA, USA), imported into Stata version 15 (Statacorp Texas; College Station, TX, USA), and cleaned. We first computed descriptive statistics and conducted cross-tabulations for sources of fruits and vegetables and handling and processing measures per stage along the chain. We also conducted trend tests (LIST TEST) to evaluate changes in pesticide concentration along the supply chain.

### 2.2. Qualitative Methods

We conducted in-depth interviews (IDI) with stakeholders with the aim of understanding in detail the handling and processing practices that may influence the concentration of pesticide residues from farm to fork. Sixteen (16) stakeholders were interviewed including farmers (4), transporters (4), market vendors (4) and consumers (4). The stakeholders interviewed were selected from the 186 that had participated in the semi-structured questionnaire interviews. The IDIs were carried out using a guide based on the existing literature focusing on post-harvest handling and processing methods [15,27,28]. The IDI guide was developed by a public health specialist [C.S.] and reviewed by public health specialists (D.G., J.C.S. and A.B.) with over 20 years’ experience in conducting research before the final version was agreed upon. Prior to data collection, the IDI guide was also pretested in the Jinja district and modified based on feedback. The IDIs were conducted face-to-face by a public health specialist. All interview proceedings were recorded using two electronic tape recorders working simultaneously to avoid loss of information. A hand-written journal was also kept by C.S.

In-depth interviews were transcribed verbatim and translated to English. The transcripts were reviewed by C.S. to generate meaning units and shared to D.G., J.C.S. and A.B. for review. All IDIs were imported into NVivo qualitative data analysis software; QSR International Pty Ltd. Version 12 and coded. Data were analyzed using conventional content analysis. During analysis, several emerging ideas were identified from the narratives based on the words and phrases that appear more often as organized codes by NVivo software. The emerging ideas were organized into categories based on recurring ideas to form sub-themes. The sub-themes were merged to identify overarching themes, which were described and presented to support quantitative data with quotes from the interviews.

## 3. Results

### 3.1. Sociodemographic Characteristics of Farmers

As noted above, sociodemographic characteristics were collected only from the farmers. The majority 42/50 (82.0%) of the participants were male and married 40/50 (80.0%). Most of the farmers had attained either primary 20/50 (40.0%) or secondary 21/50 (42.0%) education and were selected from Wakiso district 29/50 (58.0%). The average age of farmers was 43.0 years (standard deviation = ±13.6 years) (Table 1).

### 3.2. Trends of Pesticide Residue Concentrations along the Supply Chain (Farm-to-Fork)

From an earlier study [22], pesticides that met the assumption for linearity along the supply chain and were evaluated for significance are shown in Appendix A. A decreasing trend was observed for dioxacarb, while increasing trends were observed for methidathion and quinalphos after the trend test. No trends were observed for the remaining pesticide residues, but they were commonly detected (Table 2).

When discussed with stakeholders along the chain, qualitative findings revealed that pesticide use on fruits and vegetables occurred at all stages from farm to fork (Table 3). Farmers mentioned that they spray fruits and vegetables with pesticides to protect them from pests and disease as soon as they plant them, but also increase the frequency of spraying when they start to ripen to improve the quality of their products.

*“When the tomatoes are starting to get ripe, we don’t stop spraying because pesticides keep the tomatoes from being attacked by pests. If I am going to harvest tomorrow, I spray today because pests are not interested in the ripe tomatoes but we have to keep the unripe ones safe*”.(IDI—Farmer)

Several farmers did not observe the pre-harvest intervals while applying pesticides to fruits and vegetables. Many farmers did not know the pre-harvest interval (PHI) for the pesticides they were using and the importance of the PHI to the health and safety of consumers. Market vendors and consumers also mentioned a misconception that fruits and vegetables with pesticide residues are fresh compared with those without pesticides. Farmers and market vendors also apply pesticides to increase their shelf life on the market.

“*Sometimes the fruits and vegetables are sprayed during harvesting. Some clients do not buy fruits and vegetables without visible pesticide residues, claiming that they have a short shelf life*”. (IDI—Transporter)

### 3.3. Fruits and Vegetable Movement from Farm to Fork along the Supply Chain

Out of the 50 farmers visited, 49 had sold their fruits and vegetables to transporters (34; 68%), market vendors (11; 22%) and homes or restaurants (4; 8%). Market vendors dominated the second and third stages of the chain buying either directly from the farm (11; 22%) or larger markets (43; 95.6%). Whereas some consumers (Home/Restaurant) bought their fruits from the farm, the majority (42; 93.3%) purchased from a market vendor and transporter (Figure 1).

Consumers said that they typically bought from large or community markets, transporters (suppliers), farms, as well as roadside vendors/hawkers.

“*I buy fruits and vegetables from the markets around when I am buying in small quantities but our large volume supplies come from Owino market*”. (IDI—Consumer)

While most of the transporters reported buying fruits and vegetables from farmers, market vendors indicated farms and large and community markets as their source of produce.

“*I buy fruits and vegetables from different farmers in the sub-counties of Nama, Kimenyedde, Nagojje and others*”. (IDI—Market Vendor)

Stakeholders along the chain relied on diverse means to transport fruits and vegetables along the chain. The majority relied on public transport (taxi) and motorcycles to transport fruits and vegetables from once stage and another. A few mentioned bicycles as the vehicle used to transport their fruits and vegetables.

“*They are brought on public transport (taxi). When they are returning from Kampala, they load the fruits and vegetables in sacks and deliver them to me here*”. (IDI—Consumer)

### 3.4. Handling and Processing Methods That Influence the Trends of Pesticide Residues in Fruit and Vegetable from Farm to Fork

Handling and processing practices are summarized in Table 4. Most consumers indicated that they washed fruits with water before cooking. The majority of the farmers, transporters and market vendors reported that they did nothing to the fruits and vegetables. A few mentioned peeling of the outer layers as a handling and processing method that they practiced on fruits and vegetables (Table 2).

Most consumers mentioned washing fruits and vegetables with water as their common practice before cooking or blending. Some consumers also said they use warm water while others used water and a sponge to scrub the outer surface, especially for watermelons. Others mentioned they use a cloth to wipe soil or dirt that could have accumulated on the fruit or vegetable, especially if it rained towards harvesting.

“*When COVID-19 had just started, we would use warm water to wash the fruits but we have now stopped. We use normal tap water to wash the fruits*”.(IDI—Consumer)

Other stakeholders along the chain did not wash the fruits and vegetables, citing that consumers prefer them with visible pesticide residues (whitish coloring). Others also said that washing moisturizes the fruits and vegetables, increasing their chances of rotting and, consequently, reducing shelf life on the market.

“*I don’t wash. I was told by a farmer that washing reduces fruits and vegetables’ shelf life*”.(IDI—Market vendor)

Unlike the quantitative findings, some consumers revealed that they wash their fruits and vegetables with sodium carbonate before preparation to reduce the concentration of pesticide residues on them.

“*When I am preparing my own fruits for consumption, I wash them with sodium bicarbonate. I consume lots of fruits as snacks or juice. Before preparation, I use the sodium bicarbonate to wash them*”. (IDI—Consumer)

Stakeholders also said that they peel the outer layer of the fruit or vegetable, especially tomatoes, egg plants and cabbages. They explained that peeling of the outer layer helps reduce the pesticide residues on a particular fruit or vegetable. Blending fruits to make juice was another processing method that was used by consumers.

“*I usually peel the outer layer off of the tomatoes because it helps reduce the residue. I do the same for eggplants*”.(IDI—Consumer)

While quantitative results showed that pesticides are not applied along the chain, some market vendors revealed that they at times spray fruits and vegetables with pesticides, especially if they do not show any sign of residues on them. They explained that this is done to ensure that their tomatoes last long on the market without rotting.

“*If I get tomatoes that have not been sprayed, I buy the pesticide (mancozeb) and spray them to increase their shelf-life. But, I do not spray the watermelons because they are all bought within two or three days*”. (IDI—Market Vendor)

Stakeholders along the chain reported handling procedures aimed at maintaining the quality of fruits and vegetables. During transportation, stakeholders mentioned sacks, boxes, baskets and polythene bags as packaging materials used to transport fruits and vegetables from one stage to another. When asked whether the packaging material could have been used for storing pesticides, stakeholders denied using such materials.

“*I have wooden boxes, baskets and sisal and ordinary sacks. I pack the fruits and vegetables in these boxes, baskets or sacks and transport them to the market. I normally buy and transport produce the same day to the market*”.(IDI—Transporter)

Boxes, polythene liners, sacks, and refrigerators were also used during storage of fruits and vegetables. Fridges were largely used for storage by consumers. Farmers, transporters and market vendors largely used polythene liners, sacks and boxes. They also mentioned storing fruits and vegetables in dry conditions including the floors of store rooms or even outside in the open environment to ensure that they have access to adequate fresh air.

“*I store fruits and vegetables on sacks or polythene liners. I sort and spread them into basket and other containers for display. I put the remaining fruits and vegetables inside the store room*”. (IDI—Market Vendor)

“*I keep the fruits and vegetables in a store room. I put them in wooden boxes that are well aerated baskets and scatter them around the room*”. (IDI—Market Vendor)

## 4. Discussion

This mixed methods study examined trends of pesticide residue concentrations from farm to fork. We also examined handling and processing practices along the food chain. Our findings demonstrate the complexity of the food supply chain, but also present opportunities for reducing pesticide residue along the chain. Importantly, changing stakeholder perspectives about the importance of pesticide residues on food could increase food safety. Understanding these factors is important to promote strategies to reduce pesticide residues on fruits and vegetables and consequent human exposures, especially to vulnerable populations.

While the majority of consumers purchased produce at the terminus of the farm to fork chain, some bought food directly from farmers and transporters. These differences may be due to the geographical location of the different stakeholders along the chain, cost, quality and amount of fruits and vegetables, as well as convenience. Unlike Europe and the United States [30,31], Uganda and some sub-Saharan African countries do not have a policy requiring identification of the origin of certain foods on the market, yet many foods are contaminated or adulterated along the chain. This makes tracing of the source of the contamination difficult or impossible. The lack of a regulated food supply chain has led to less knowledge about food handling strategies that minimize pesticide use, residues and exposure while maintaining good food hygiene and safety standards.

Pesticide residues were detected in fruits and vegetables at all stages along the chain. We found that some pesticides decreased while others increased. Various studies from Ghana and Belgium have highlighted similar trends in food contaminants from farm to fork [32,33,34]. Our qualitative findings also highlighted that fruits and vegetables at all stages of the chain are likely to contain pesticide residues. These residues may in part be due to farmers’ failure to observe the pre-harvest interval between pesticide application and harvesting [35]. Some stakeholders also apply pesticides along the chain to increase the shelf life of fruits and vegetables. In fact, some market vendors reported that they apply pesticides, especially on tomatoes, so visible residues would appeal to consumers. Handling methods including washing, peeling, cooking, blending and drying and other practices have also been shown to increase or decrease the concentration of pesticide residues in fruits and vegetables [16,26,27,29,36]. The use of pesticide in the prevention of vector- and vermin-related diseases and nuisances could also be responsible for some contamination in the produce [37]. Our study highlights the need for farmers and other stakeholders along the chain to ensure the hygiene and safety of the fruits and vegetables on the market by observing the pre-harvest interval for pesticide application before harvesting and proper handling and processing throughout the food chain.

General food handling and processing by consumers, while not necessarily aimed towards a reduction in pesticide residues, may result in lower exposures at the point of consumption. For example, previous studies have shown that food processing reduces pesticide residues in food [15,26,27,29,36,38]. Among the processing methods, peeling has been considered the most effective [36,39], while washing the least effective, at reducing pesticide concentration in produce [27,28]. Washing removes pesticide residues that are able to be dislodged and attached to the surface, while peeling also removes pesticides that penetrate the fruit and/or vegetable cuticle. Washing fruits and vegetables can actually increase the concentration of pesticide residues in food if it is carried out using water that is contaminated [40]. Washing fruits with cleaning agents and juicing/blending has also been reported to increase pesticide concentration in fruits and vegetables [15,16,28].

Treatment of fruits and vegetables to increase shelf life and reduce loss across the supply chain is potentially a significant source or dietary pesticide exposure. Several studies highlight physical (heating, irradiation and edible coating), chemical (antimicrobial, antioxidants and anti-browning agents, nitric acid and sulfur dioxide), and gaseous (ozone, controlled atmospheric storage and ethylene) [12] methods as safe post-harvest treatment strategies for fresh produce. While these modern treatment methods ensure the hygiene and safety of fruits and vegetables, they are expensive and may not be affordable to many farmers in Uganda, especially the smallholder farmers that were involved in our study. As a result, some farmers and market vendors have resorted to using pesticide as a treatment to increase the shelf life of fruits and vegetables along the food chain, as demonstrated in our study. This increases the potential of dietary pesticide exposures and health risks among consumers.

This study has several limitations. We did not track specific fruits and vegetables from the farm to the fork and had small samples at every stage. The supply chain is very complicated with several players at almost every level. Transporters buy fruits and vegetables from several farmers in order to meet their demand but also supply to several markets. Tracking a specific fruit and/or vegetable along the chain would be very costly, so we conducted sampling at different stages. Due to this approach, the difference in pesticide residue concentration along the chain might not reflect practices of individual farmers and time for pesticide degradation. A comprehensive study that can track specific fruits and vegetables from farm to fork with larger samples collected per stage could address this limitation. Additionally, some of the fruits and vegetables of interest to us were not available in the study areas due to seasonal or other factors. Third, stakeholders were asked to report on their practices along the chain including whether they had applied pesticide to fruits and vegetables post-harvest. Given that this use may be considered inappropriate, some stakeholders might have misreported pesticide use, despite assurances of confidentiality and, thus, introduced information bias. Strengths of the study included that the study area represents a large proportion of the Ugandan population and many different commonly eaten foods. This made it possible to capture different practices at different stages of the chain and examine how they affect the hygiene and safety of fruits and vegetables. We also used laboratory testing to measure a wide variety of pesticides in the produce. Stakeholders were also traced down from farm to fork through phone calls, giving a relatively representative account of practices along the chain. Finally, the use of mixed-methods methodology helped to identify some practices during the in-depth interviews that were not raised in the semi-structured questionnaire interviews.

## 5. Conclusions

This is one of the first studies to investigate trends of pesticide residues and movement patterns along the food chain from farm to fork in Uganda. It provides important information on the determinants of pesticide residues along the food chain and potential opportunities for reducing their concentrations. Overall, fruits and vegetables moved from the farm through transporters and market vendors to final consumers, with some consumers buying directly from the farmers or transporters. Pesticide residues declined along the chain for dioxacarb but some increased, as demonstrated by methidathion and quinalphos. From the qualitative findings, it is clear that the concentration of pesticide residues on fruits and vegetables may be in part attributed to farmers’ pesticide application practices near harvest. Other reasons why pesticide residues increase or decrease on fruits and vegetables along the chain include handling and processing practices used by stakeholders such as washing, peeling, spraying with pesticides, blending and wiping with a cloth. Education is needed to train farmers on good agricultural practices, especially the importance of observing pre-harvest intervals.

## Figures and Tables

**Figure 1 ijerph-19-01350-f001:**
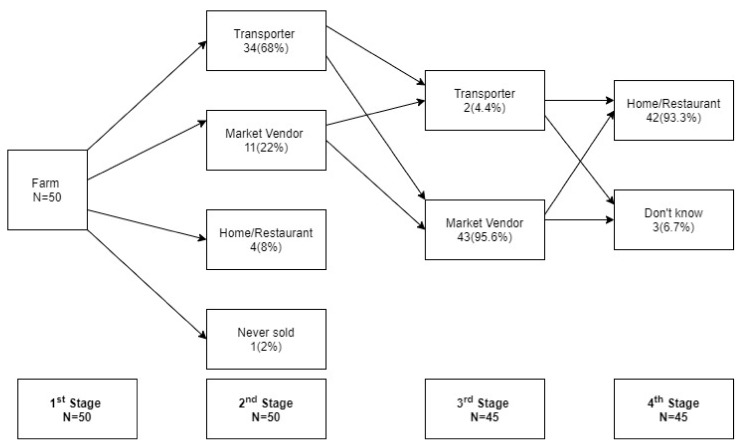
Fruit and vegetable movement along the supply chain from farm to fork.

**Table 1 ijerph-19-01350-t001:** Sociodemographic characteristics of farmers.

Variable	Frequency *n* = 50	Percentage (%)
**Sex**		
Male	41	82.0
Female	9	18.0
Age mean (±SD)	43.0 (±13.6)	
**Age category**		
18–35	18	36.0
36–49	16	32.0
50–81	16	32.0
**Religion**		
Christian	40	80.0
Moslem	10	20.0
**Education Level**		
Tertiary	4	8.0
Secondary	21	42.0
Primary	20	40.0
None	5	10.0
**Marital status**		
Married	40	80.0
Single	6	12.0
Widowed/divorced	4	8.0
**District**		
Wakiso	29	58.0
Mukono	21	42.0

**Table 2 ijerph-19-01350-t002:** Pesticide residue concentration trends in fruit and vegetable from farm to farm along the supply chain in ng/kg.

Pesticide Residue	LOD (ng/kg)	Farm *n* = 50	Market *n* = 50	Street *n* = 20	Restaurant *n* = 20	Home *n* = 20	*p*-Value
		Mean ± SD	Mean ± SD	Mean ± SD	Mean ± SD	Mean ± SD	
Mevinphos	33.9	25.7 ± 129.0	34.9 ± 147.9	3.7 ± 14.3	3.6 ± 12.4	14.4 ± 64.3	0.20
Dichlorvos	15.3	6448.5 ± 22,252.2	126.5 ± 387.4	56.1 ± 147.7	265.8 ± 960.5	504.9 ± 1128.8	0.81
Profenofos	9.7	18,196.7 ± 75,391.2	7106.6 ± 37,302.2	6328.9 ± 19,767.4	10,879.2 ± 29,572.2	722.2 ± 2570.4	0.75
Methomyl	33.5	35.8 ± 84.8	52.9 ± 117.0	7.1 ± 26.9	39.2 ± 103.6	4.6 ± 20.1	0.14
Dioxacarb	13.1	5273.9 ± 17,656.2	4068.9 ± 15,284.1	3462 ± 15,484.2	0	0	0.01 *
Methiocarb	43.9	28.2 ± 74.7	13.2 ± 37.9	25.9 ± 48.0	6.9 ± 18.9	1.3 ± 3.1	0.61
Acetamiprid	20.4	6626.5 ± 24,394.5	3561.2 ± 11,552.9	2349.4 ± 4791.9	2235.2 ± 5549.1	810.6 ± 2518.5	0.91
Bifenthrin	18.7	113.3 ± 392.4	258.3 ± 1018.4	7.5 ± 33.6	26.8 ± 68.7	4.2 ± 13.1	0.11
Benfuracarb	50.0	37,668.5 ± 175,456.4	8207.5 ± 29,972.8	2802 ± 5206.2	8.1 ± 2.6	3146.6 ± 9950.6	0.71
Lambda-Cyhalothrin	21.1	167.7 ± 389.7	189.4 ± 423.8	198.6 ± 353.2	126.1 ± 234.4	149.9 ± 242.2	0.45
Cypermethrin	11.1	736.7 ± 2883.3	314.0 ± 1122.9	808.4 ± 2792.7	172.3 ± 493.9	100.1 ± 348.2	0.92
Spirotetramat	18.4	45.8 ± 192.0	25.2 ± 97.3	39.1 ± 92.9	5.1 ± 11.2	15.6 ± 28.8	0.90
Flufenoxuron	15.4	5.8 ± 26.6	3.5 ± 13.0	0.3 ± 1.6	2.2 ± 7.9	0.9 ± 4.1	0.49
Proquinazid	14.6	540.2 ± 1767.2	121.4 ± 606.8	427.0 ± 1350.4	168.7 ± 533.5	0	0.41
Methidathion	14.4	0	0	1.9 ± 6.0	0	39.8 ± 110.9	0.01 *
Carbaryl	7.7	4.3 ± 27.6	4.2 ± 49.9	2.9 ± 12.8	0	15.5 ± 49.9	0.94
Azoxystrobin	7.4	3356.1 ± 11,900.6	0	0	2472.4 ± 5931.1	6652.6 ± 21,037.3	0.35
Fenarimol	13.1	347.5 ± 1092.0	139.7 ± 612.9	255.5 ± 659.7	382.9 ± 1051.6	942.4 ± 2494.6	0.88
Isofenphosmethyl	20.0	244.8 ± 611.0	58.3 ± 219.5	271.3 ± 857.9	61.3 ± 193.8	1063.5 ± 1743.2	0.31
Ethoprophos	84.6	0	16.5 ± 82.7	0	15.5 ± 49.1	44.4 ± 140.5	0.11
Quinalphos	31.8	19.1 ± 68.2	80.5 ± 191.3	59.0 ± 186.7	195.2 ± 483.4	119.7 ± 174.7	0.01 *

LOD—limit of Detection, SD—standard deviation, * *p* ≤ 0.05.

**Table 3 ijerph-19-01350-t003:** Pesticide use practices, movement patterns and handling and process practice of fruits and vegetables from farm to fork.

Theme	Sub-Themes	Basic Themes
Fruits and vegetables contain pesticide residues	Pre-harvest treatment	Spray with pesticides
	Good agricultural practices	Pre-harvest period
	Increase shelf life	Spray with pesticides Prefer fruits and vegetables with residues
	Condition of fruits and vegetables	Pesticide residues
		Customer preference
Movement patterns of fruits and vegetables along the chain	Sources of fruits and vegetables	Large markets Community markets Suppliers (Transporters) Roadside seller/Hawkers Farm
	Transportation means	Public transport (Taxi) Motorcycle Bicycle
Handling and processing practices for fruits and vegetables along the chain	Packaging during transportation	Sacks Boxes Boxes/baskets/sacks Polythene bags
	Storage	Wooden boxes/baskets Polythene liners/Sacks Outside Fridge Food store on the floor Dry conditions
	Processing	Washing with water Washing with chemicals Peel outer layer Blending Spray with pesticides Wiping with a cloth

**Table 4 ijerph-19-01350-t004:** Handling and processing methods used by stakeholders along the chain from farm to fork *n* = 50.

Variable	Farm	Transporter	Market Vendor	Consumer
	*n* (%)	*n* (%)	*n* (%)	*n* (%)
Wash with water	2 (4)	NR	5 (10.0)	41 (82.0)
Wash with chemical	NR	NR	NR	NR
Peel off outer layer	5 (10)	3 (6.0)	3 (6.0)	10 (20.0)
Spray with pesticide	00 (00.0)	00 (00.0)	00 (00.0)	NR
Sun drying	NR	NR	NR	NR
Cooking	NR	NR	NR	21 (42.0)
Boiling	NR	NR	NR	4 (8.0)
Oven drying	NR	NR	NR	NR
Nothing	43 (86)	28 (56.0%)	36 (72.0)	NR

NR = behavior not reported.

## Data Availability

Due to ethical restrictions related to protecting patient privacy, data cannot be made publicly available. Data are available upon request from the corresponding author.

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
