# Peer review of "Pesticide Residue Trends in Fruits and Vegetables from Farm to Fork in Kampala Metropolitan Area, Uganda—A Mixed Methods Study"

_ijerph, 2022, doi:10.3390/ijerph19031350_

Round 1

Reviewer 1 Report

The paper takes on a relatively unexplored subject and uses multiple methods to approach the question of pesticide residues in fruits and vegetables in a region and country that faces serious environmental and food safety challenges. As such, it deserves consideration for publication. 

The introduction makes several observations that are not supported by cited references and presuppose the outcomes of the research. Before it can be published, these need to be corrected. Specifically:

Lines 43-44 "Some farmers do not comply with the recommended pre-harvest interval . . ." What is the source of this statement? How many farmers were known to be non-compliant prior to the study taking place? It would be fair to state this as a hypothesis that the research intends to address, at least partly, but to put it in the introduction as a statement of fact without a source is inappropriate.

Lines 44-46 "It has also been reported . . . " By whom? What is the source? The passive voice should not be used here. 

". . . that some stakeholders along the food chain . . ." I would not call them "stakeholders". That is too vague and possibly inaccurate. Are you talking about brokers, handlers, distributors? Taxi drivers? Consumers? Be specific and cite the sources. If there are no sources that can be cited prior to the study, again this is presupposing the results of the research and is inappropriate.

Line 73: Escherichia

Lines 98-101: Explain how subjects were selected. 

Lines 199, 203, and elsewhere: Again "stakeholder" may be inaccurate. Better to call them "interviewees" or "subjects". 

Table 2 is difficult to read. The n numbers are not reported. The wide variation between sites in the supply chain and relatively large standard deviations suggest to me that the sample sizes are too small. In addition to the Limit of Detection, the Maximum Residue Levels (MRLs) should also be reported. How many samples were above the MRLs? These may bear discussion and follow-up, particularly in light of uses that don't follow label instructions.

Supplementary Table 2 helps some. I would consider breaking Table 2 into 2-3 tables and add data from the Supplementary Table.

Line numbering stopped after Table 2, so the rest of the comments refer to sections.

Section 3.3: The one farmer who never sold anything seems to be an outlier or irrelevant. 

Section 3.4 "Eggplant" is one word. 

The various quotes seem disjointed. In some cases, I think these might be better reported in the supplementary file. 

Section 4: "The lack of a functional food supply chain tracking system has led to infiltration of middlemen who do not comply with foods hygeine and safety standards."

This is a problematic sentence on several levels. Is the supply chain not functional, or is it the tracking system? Is the word "infiltrated" correct? It almost sounds like being a middleman is a crime or act of treason. 

Again, third-party documentation of their non-compliance would help back up your case that this is not just your research. You are not the police or a food inspection agency. As researchers, you can only say that they appear to be non-compliant until the proper authorities have issued them the non-compliance. 

"Food hygeine" is without an 's".

The study's reported limitations should include some the issues I identified above, including the small sample sizes and potential selection biases.

I would like to see some discussion of what can be done to reduce pesticide residue levels. 

Another thing not addressed anywhere is that there is a difference between exposure and risk. Pesticides toxicity varies widely and dietary risks are not the same for all pesticides. Perhaps it is too late to analyze what the relative risks are of each of the active ingredients found in the samples, but the discussion should at least acknowledge this and possibly consider further study to look at which specific pesticides pose the greatest risks. 

The conclusions can be made more focused. 

Reviewer 2 Report

My comments are pointed out in the attached PDF document.

Future suggestion: investigate the correlation between pesticides and the incidence of diseases like cancer in the local people.
